# Methodology to standardize heterogeneous statistical data presentations for combining time-to-event oncologic outcomes

**April E. Hebert**[1], **Usha S. Kreaden**[1], **Ana Yankovsky**[1], **Dongjing Guo**[1], **Yang Li**[1], **Shih-Hao Lee**[1], **Yuki Liu**[1], **Angela B. Soito**[2], **Samira Massachi**[3], **April E. Slee**[4]*

**1** Intuitive Surgical, Sunnyvale, California, United States of America, **2** ABS Consulting LLC, Oakland, California, United States of America, **3** Stratevi, Santa Monica, California, United States of America, **4** New Arch Consulting, Issaquah, Washington, United States of America

* april.slee.15@ucl.ac.uk

**Data Availability Statement:** All relevant data are within the paper and its Supporting information files.

## Abstract

Survival analysis following oncological treatments require specific analysis techniques to account for data considerations, such as failure to observe the time of event, patient withdrawal, loss to follow-up, and differential follow up. These techniques can include Kaplan-Meier and Cox proportional hazard analyses. However, studies do not always report overall survival (OS), disease-free survival (DFS), or cancer recurrence using hazard ratios, making the synthesis of such oncologic outcomes difficult. We propose a hierarchical utilization of methods to extract or estimate the hazard ratio to standardize time-to-event outcomes so that study inclusion into meta-analyses can be maximized. We also provide proof-of concept results from a statistical analysis that compares OS, DFS, and cancer recurrence for robotic surgery to open and non-robotic minimally invasive surgery. In our example, use of the proposed methodology would allow for the increase in data inclusion from 108 hazard ratios reported to 240 hazard ratios reported or estimated, resulting in an increase of 122%. While there are publications summarizing the motivation for these analyses, and comprehensive papers describing strategies to obtain estimates from published time-dependent analyses, we are not aware of a manuscript that describes a prospective framework for an analysis of this scale focusing on the inclusion of a maximum number of publications reporting on long-term oncologic outcomes incorporating various presentations of statistical data.

## Introduction

Acute short-term outcome trials comparing robotic-assisted surgery with open surgery and non-robotic minimally invasive surgery have been evaluated for decades. Many publications have reported short-term outcomes such as length of stay and complication rates associated with robotic-assisted surgery compared to traditional open surgery and non-robotic minimally invasive surgery [1, 2]. For cancer-related surgical procedures, a comprehensive comparison of robotic-assisted surgery to traditional open or non-robotic minimally invasive alternatives should evaluate mid- to long-term oncologic outcomes in addition to these acute surgical

**Funding:** All authors are employees of Intuitive Surgical, Inc. (AEH, USK, AY, DG, YLi, SL, YLiu) or consult for Intuitive Surgical, Inc. (ABS, SM, AES) and all work related to this paper was performed in the course of usual work operations and was not based on a grant or other study-specific funding award.

**Competing interests:** "All authors are employees of Intuitive Surgical, Inc. (AEH, USK, AY, DG, YLi, SL, YLiu) or consult for Intuitive Surgical, Inc. (ABS, SM, AES). There are no patents, products in development or marketed products to declare. This does not alter our adherence to PLOS ONE policies on sharing data and materials."

measures. These oncologic outcomes include overall survival (OS), disease-free survival (DFS) and cancer recurrence.

In contrast to acute surgical outcomes, where complete patient accounting is feasible, these oncologic outcomes are time-dependent and the event status for each patient is not always known. Both the occurrence of these outcomes and the elapsed time from surgery contribute to valid comparisons across surgical approaches. As such, these outcomes require specific techniques that are not necessary for comparisons of counts or continuous endpoints.

Specifically, for short-term outcomes, it is possible to analyze the presence or absence of specific events as binary endpoints and to focus only on the proportion of patients for whom the endpoint has occurred. These analyses assume that the event status is known for each patient, which is reasonable when the outcomes are measured prior to hospital discharge or within a short time after discharge. For time-dependent outcomes such as all-cause mortality, the "endpoint" is the elapsed time from surgery to death. However, it is unlikely that every death among patients in the study would be observed during the follow-up period, so for many patients, the time of death is considered missing data. Failure to observe the time of death, or censoring, can occur for reasons other than insufficient observation time. Other common reasons that an event was not observed during a study include patient withdrawal, a competing risk event (an event that precludes observing the outcome of interest), and loss to follow-up.

Fortunately, survival analysis techniques can account for failure to observe events and produce valid statistical comparisons. The single event component—time from surgery to death—is replaced by 2 components. For patients with observed deaths (or other events), these components are a binary variable indicating that the death was observed and the time from surgery to death. For patients who did not die prior to loss to follow-up, withdrawal or study conclusion, the corresponding components are a binary variable indicating that death was not observed and the last time that the patient was known to be alive (or that the event of interest had not yet happened). This framework allows adjustment for differences in follow-up time and other reasons that the follow-up data are incomplete. Other cancer-related endpoints such as recurrence and disease-free survival can be defined as similar pairs of indicators of event occurrence and the time the event occurred or the last time when the patient was known to be event-free.

For oncologic meta-analyses, the follow-up period may differ considerably across studies, and longer observation time increases the chances of observing an endpoint event. This problem is compounded for retrospective comparisons, where the follow-up period may differ considerably *within* studies. More time has elapsed since the procedure for older surgical approaches compared to newer ones, resulting in longer amounts of follow-up and more opportunity to observe events. Thus, an analysis based on event counts with no correction for differences in elapsed time will favor the newer technique even if the true event rates are similar. While statistical techniques such as propensity score matching or covariate adjustment can reduce differences in patient characteristics within studies, these methods do not address differential follow-up.

Analysis of the proportion of patients with an event at a specific time-point such as 5-year survival appears to address this problem as follow-up time is fixed across treatment groups (and studies), but this approach does not address censoring and, in general, these comparisons are not amenable to synthesis for long-term outcomes. Analysis of a single time-point can misrepresent the overall treatment effect, and this is especially true when points of maximum or minimum difference between survival curves are selected for presentation [3, 4]. Additionally, the requirement of a common time-point for meta-analyses will reduce the number of studies that can be included, and the generalizability of the findings may suffer as

a result. There are several publications further detailing the need for time-dependent analyses [5, 6].

The most common recommendation for the analysis of time-to-event outcomes is to summarize intervention effects using the hazard ratio (HR) [7]. The hazard ratio compares the instantaneous risk of events across interventions. This statistic is most interpretable when the ratio of these risks is relatively constant over the follow-up period (the "proportional hazards assumption"), but it summarizes the overall reduction in event risk for one group compared to another group even when the ratio of risks is not constant over time [8]. Hazard ratios are similar in interpretation to relative risks, but they account for time and censoring in addition to the number of events. On the natural logarithm scale (Ln), the hazard ratio is linear [7, 9] and can be aggregated using the generic inverse-variance methods [8] used in meta-analysis for other statistical measures.

Unfortunately, the hazard ratio is not always reported in publications. Conversely, other publications report multiple hazard ratios calculated from raw data, matched data, and statistical models. Comprehensive papers describing strategies to obtain estimates for time-dependent analyses [4, 8] have been previously published, but these reports lack the organizational framework to implement a large meta-analysis involving multiple teams of data extractors. The goal in our framework development was to maximize the number of included studies while limiting bias, to provide clear guidelines, and improve agreement in dual data extraction of each individual manuscript. Thus, we used a hierarchical decision tree to allow data extractors to identify the most appropriate hazard ratio to extract, or the most appropriate data and method to estimate the hazard ratio when it was not provided directly.

The purpose of this manuscript is to describe and illustrate the development of an approach for performing systematic literature reviews and meta-analyses to summarize important time-dependent oncologic endpoints. To illustrate our approach, we describe a project comparing robotic-assisted surgery using the da Vinci surgical system to open and non-robotic minimally invasive surgical procedures (PROSPERO database CRD42021240519, analysis in progress).

## Materials and methods

### Methods to extract hazard ratios

We created a hierarchical decision tree using 4 methods to extract the hazard ratios and variance estimates from the publications identified for this project. Method 1 used the direct estimate of the hazard ratio when available, while Methods 2–4 used "indirect methods" to estimate the hazard ratio from available data [6]. The decision tree shown in Fig 1 was created based on the increasing number and decreasing plausibility of required assumptions and was used to determine which technique to use to extract or estimate the hazard ratios.

The individual Methods 1–4 are described in detail below. Methods 1 through 3 have previously been described [4, 8, 10] and are recommended by the Cochrane Handbook [7]. Our base assumption was that hazard ratios are a valid comparison of overall risk between groups in directionality and magnitude even when the hazards are not proportional, but statements quantifying the comparisons (e.g., a 5 x higher risk) should not be made in the case of non-proportionality. Our main rules were that 1) all available data, outcome definitions, and stated conclusions were utilized to determine the most valid data, method, and p-value to use, and to check the accuracy of Method 2–4 calculations, 2) when there was a judgement call needed, we selected the method that was the most conservative (most disfavored) for the cohort of interest.

**Method 1.**    Since the goal of this analysis was to combine hazard ratios, reported hazard ratios were used as the first choice when available. Hazard ratios were typically presented with

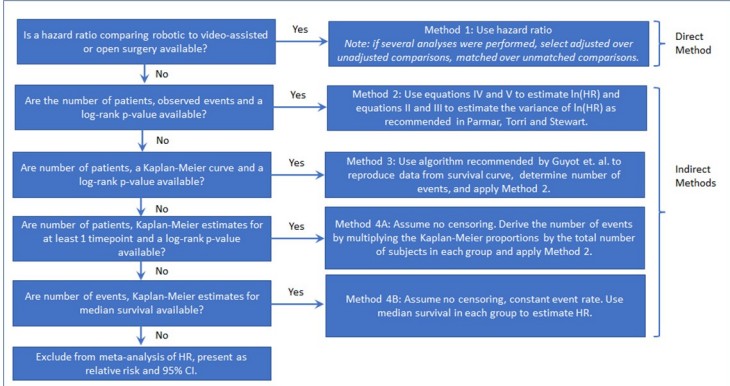

**Fig 1. Decision tree for hazard ratio extraction: Flow chart to determine which hazard ratio estimate to use based on data provided in manuscript.**

95% confidence intervals (CI), and so variances could be obtained from confidence intervals [8]:

$$Variance\ Ln(HR) = \left[ \frac{Upper\ 95\%\ CI - Lower\ 95\%\ CI}{2\Phi^{-1}\left(1 - \frac{\alpha}{2}\right)} \right]^2 \qquad\qquad (I)$$

Where "Ln" denotes the natural logarithm ($\log_e$), HR denotes the hazard ratio, CI denotes the confidence interval, $\Phi^{-1}$ is the inverse of the standard normal distribution, and $\alpha$ is the Type 1 error rate.

If only the hazard ratio and p-value are reported, the confidence interval can be calculated as follows:

$$Standard\ Error\ Ln(HR) = \left[ \frac{Ln(HR)}{\Phi^{-1}\left(1 - \frac{p\text{-}value}{2}\right)} \right] \qquad\qquad (II)$$

Where "Ln" denotes the natural logarithm ($\log_e$), HR denotes the hazard ratio, and $\Phi^{-1}$ is the inverse of the standard normal distribution. The denominator of the ratio in equation II has the form of a z-statistic on the log scale, so equation II is essentially SE = Estimate / z. Note that the standard error cannot be negative, so the calculation should use either $\Phi^{-1}$(p-value/2) or $\Phi^{-1}$(1—p-value/2) as necessary to make the resulting standard error in equation II positive. There are related formulations of this equation that produce nearly identical results in most circumstances [11]. Once the standard error is obtained, a 95% CI for Ln(HR) is given by:

$$95\%\ \text{Confidence Interval for Ln(HR)} = \text{Ln(HR)} \pm 1.96 * \text{SE.} \qquad\qquad (III)$$

Where "Ln" denotes the natural logarithm ($\log_e$), HR denotes the hazard ratio, and SE denotes the standard error. A confidence interval for HR can be obtained by exponentiating the end-points:

$$95\%\ \text{Confidence Interval} = e^{(\text{Ln(HR)} \pm 1.96 * \sqrt{(\text{variance Ln(HR)})})} \qquad\qquad (IV)$$

Where "Ln" denotes the natural logarithm ($\log_e$) and HR denotes the hazard ratio.

When multiple hazard ratios were reported, the statistical analysis that produced the hazard ratio was also captured (i.e., univariable, multivariable) to follow the extraction priority. It is important to determine an extraction preference *a priori* for when more than one hazard ratio

is reported. Our criterion was to prioritize adjusted or matched analyses over unadjusted data, and when both adjusted and matched analyses were available, to maximize group size, because analyses using entire populations account for the relative frequency of case types, severity of disease, surgeon experience, etc. and the results are more generalizable. For these reasons, we prioritized an adjusted HR using the largest sample that adequately addresses confounding (ie. adjusted analysis using whole patient population over a matched patient cohort when matching decreased the sample size). It is also crucial to convert the reference group for all HRs to the same surgical approach to perform pooled analysis. We converted all HRs so that the robotic cohort was the comparison group (i.e., not the reference group). The conversion was performed by inverting the hazard ratio and its 95% confidence interval: 1/HR [1/high CI, 1/low CI] where HR denotes the hazard ratio and CI denotes the 95% confidence interval. To identify the reference group when it was not explicitly stated, the author conclusions, event n, KM curves, and survival rates were used along with the guide shown in Table 1.

**Method 2.** When hazard ratios were not available, the hazard ratio was estimated using log-rank analysis statistics. This method estimates the variance and hazard ratio using the number of patients and events in each arm, and the log-rank p-value [8].

$$V_r = \frac{O_{Total} R_r R_c}{(R_r + R_c)^2} \tag{V}$$

$$Variance\ Ln(HR) = \frac{1}{V_r} \tag{VI}$$

$$O_r - E_r = \frac{\sqrt{O_{Total} \times R_r \times R_c}}{(R_r + R_c)} \times \Phi^{-1}\left(1 - \frac{p}{2}\right) \tag{VII}$$

$$Ln(HR) = \left(\frac{O_r - E_r}{V_r}\right) \tag{VIII}$$

For the above equations, $V_r$ is the inverse variance of the log hazard ratio for the robotic group, $O_{Total}$ is the total number of events in the robotic plus the comparison group, and $R_r$ and $R_c$ are the number of patients in the robotic and comparison groups, respectively. The "Ln" denotes the natural logarithm ($\log_e$) and HR denotes the hazard ratio. The $O_r$ and $E_r$ are the number of observed and expected events in the robotic group, respectively, $\Phi^{-1}$ is the inverse of the standard normal distribution, and the p-value is assumed to be 2-sided if not otherwise stated and from the log-rank test. Because p-values were assumed to be two-sided, manual assignment of direction was adjusted by selecting the appropriate equation from these two options: $\Phi^{-1}\left(1 - \frac{p}{2}\right)$ or $\Phi^{-1}\left(\frac{p}{2}\right)$ so that the result is negative when survival in the robotic group was higher/better and positive when survival in the robotic group was lower/worse.

The p-value from the log-rank test of the Kaplan-Meier curves should be nearly identical to the log-rank p-value from an unadjusted (univariable) Cox proportional hazards model; therefore, a Kaplan-Meier log-rank p-value in conjunction with an HR, a KM curve, or a KM

**Table 1. Determination of reference group for hazard ratio: Table to assist extractors in identifying reference group, determining need to invert hazard ratios.**

| Hazard Ratio Direction | Open/Minimally Invasive Reference | Robotic Reference |
|---|---|---|
| HR > 1 | Robot is Worse | Robot is Better |
| HR < 1 | Robot is Better | Robot is Worse |

survival estimate, was used to estimate the HR with the other Methods 2–4. If the source of the p-value was not clear, we reproduced p-values from the Chi$^2$ and Fisher's Exact test using cohort totals and event counts for comparison. This step was used to reduce the possibility that either of these tests was the source of the p-value before assuming it is calculated from a log-rank test. When using a p-value to calculate an HR or CI, if it is reported as "less than" (e.g., p<0.001), check for a log-rank statistic and if reported, use it to derive a more precise p-value. If no log-rank statistic is reported, the convention is to treat it as equal (e.g., p<0.001 becomes p = 0.001). This convention has the effect of biasing the resulting estimates toward the null hypothesis.

The number of patients relates to the number of patients included in the analysis (n at risk at time zero) and may not always be equal to the total sample size. For example, it might be necessary to subtract patients with no follow up. For overall survival, the total number of deaths may be calculated by summing across causes of death (e.g., dead of other (DOO) + dead of disease (DOD)), subtracting the number of alive from the total sample size, or by calculating from a proportional death rate. For composite endpoints such as disease-free survival, it is important to be cautious to avoid double-counting patients that experienced multiple events (a patient that experiences recurrence and death would be counted twice if recurrence event n and a death event n were summed), but if there is no evidence it would be inaccurate, DOO + DOD + alive with disease (AWD), or overall mortality + (total recurrence minus DOD) equations could be used.

**Method 3.** If neither Method 1 or Method 2 could be applied based on information available in the publication, but the log-rank p-value and a Kaplan-Meier curve were available, individual patient data was reconstructed from the published Kaplan-Meier curve using the iterative algorithm recommended by Guyot et al. [10, 12, 13]. First, the Kaplan-Meier curve was digitized [14] to retrieve x (time) and y (survival) coordinates. The Guyot algorithm can be implemented using SAS or R—code and examples can be found in S1 Appendix. The Guyot algorithm makes use of the fact that the Kaplan-Meier estimator for time t is the product ($\prod$) of the probability of surviving the most recent interval ending at time t and all intervals prior to time t:

$$Survival\ to\ at\ least\ time\ t = \prod_{t_i \leq t} \left(1 - \frac{d_i}{n_i}\right) \qquad (IX)$$

Where $d_i$ and $n_i$ represent the number of deaths in interval i and the number at risk at the start of interval i, respectively. Kaplan-Meier estimates only change when an event is observed, so the timing of events is reflected by changes in the curve. The Guyot algorithm divides the follow-up period into intervals, and iteratively adjusts the distribution of censored observations and events until the survival estimate at time t is closest to the result extracted from the graph.

The Guyot algorithm was evaluated using a validation exercise that involved simulating data, exporting Kaplan-Meier estimates to be used as input to the Guyot algorithm, reconstructing the individual patient-level data (IPD) using the Guyot algorithm, and comparing the hazard ratio obtained from the original and reconstructed data. This process is shown in Fig 2.

This simulation exercise demonstrated that the hazard ratio from the Guyot reconstructed IPD was an accurate estimate of the original hazard ratio used in the simulation. We found the greatest deviations from the original hazard ratio when there were relatively few events (95% censored observations), or no information about n at risk was available. In these cases, published methods provided in Tierney et al. [4] were utilized to obtain or estimate the n at risk.

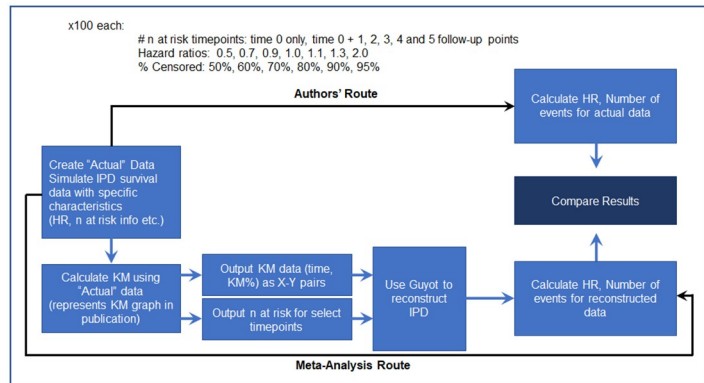

**Fig 2. Simulations to verify correct implementation of Guyot algorithm: Visual illustration of simulations explored for validation of Guyot algorithm.** Details can be found in S2 Appendix.

Once the individual patient data had been re-constructed using this algorithm, the number of events in each arm was obtained and the Method 2 estimation procedure was applied. The hazard ratio can also be calculated directly using Cox Proportional Hazards. This was done as part of the simulation study; we calculated the hazard ratio directly from the Guyot IPD and indirectly using Method 2 and compared them. We found no material difference in the resulting hazard ratios, which confirmed that our approach of using the number of events and Method 2 was valid (see results in S2 Appendix). However, similar to what others have observed, both methods are less accurate compared to the simulated value when the proportional hazards assumption does not hold [15].

The use of Method 3 is associated with additional considerations. If the number of events is few enough, manually counting them may be an option, but the result should be confirmed by manually calculating the KM estimate. The quality of the published image affects the ability to accurately count or digitize the Kaplan-Meier curve. There were a few instances where poor resolution and distortion precluded accurate extraction and the use of Method 3, so in those cases, we used Method 4 when possible. If no KM curve is shown, but the time of each event was reported along with summary information about the follow-up distribution, an approximation to the KM curve can be constructed manually. As a quality control step, we produced Kaplan-Meier graphs for each study endpoint that utilized Method 3 and compared these graphs to the original published graphs, refining the Guyot input until they matched. The log-rank p-value can be used to adjust censoring in the Guyot algorithm when n at risk over time is not reported. We also used the log-rank p-value to help determine how well the differences in the reconstructed data reflected differences in the published data. Lastly, some manuscripts included 3 cohorts (e.g., open, laparoscopic, and robotic), with a single p-value testing the overall null hypothesis that there are no differences in survival curves [16]. When the overall p-value seemed to reflect pairwise differences (assumption of similar variances), then the 3-way p-value was assumed to be a reasonable approximation of each pairwise comparison. When the overall p-value clearly did not reflect pairwise differences, but the re-constructed data matched the publication relatively well, and the overall p-value from the reconstructed data was similar to the publication, then the reconstructed data were used to calculate pairwise p-values. An additional limitation of this approach is that it may result in type 1 error inflation because the overall comparison is not equivalent to two comparisons to a control. If these criteria were not met, we used Method 4.

**Method 4.** Method 4 was used in instances when a time-dependent analysis was performed, but it was not possible to account for the censoring distribution because the reported information was limited to a single time point per cohort, either in the form of Kaplan-Meier estimates, or median survival. Method 4a: If Kaplan-Meier estimates for at least one timepoint and a log-rank p-value were reported, the number of events was estimated by multiplying the Kaplan-Meier estimate by the total number of patients to get the number of patients without events, and then subtracting that number from the cohort total to get the number of patients with an event (if failure was presented, no subtraction was performed). We determined *a priori* which timepoint would be used when several Kaplan-Meier estimates were provided. We preferentially used the latest timepoint. Method 2 was then applied to these estimated event counts. This approach assumes no censoring, that the survival curves do not cross after the estimated time point, and that the hazards are relatively proportional, but it is generally preferable to excluding the study as information about the comparisons across groups will be consistent as long as the hazards are relatively proportional over the study follow-up period and the censoring mechanism is not different across interventions. We also utilized the conclusions of the authors to determine if this approach would accurately reflect the overall comparison between cohorts, and discrepancies were cause for excluding the data if the results and conclusions conflicted and the correct result was unclear. These assumptions may or may not be reasonable and should be mentioned as a limitation when this approach is employed. We strongly recommend a sensitivity analysis excluding Method 4 to understand the possible impact of these assumptions on the overall conclusions.

Method 4b: Median survival is not often reported following curative surgical resection of cancer, especially when follow up is 5 years or less, as it requires enough deaths to reach 50% survival. However, if this is the only time-dependent analysis reported, it may be possible to estimate an HR using median survival time in each group and the number of events in each group. Using the median survival produces a reasonable estimate when the cohort sizes are similar and when there is a constant event rate. The approximation using median survival should be used with caution as it relies on exponentially distributed events (that is, not only the ratio of event rates but the underlying event rate must be approximately constant) and similar cohort sizes [17].

$$HR \approx \frac{Median\ survival\ time_{Comparison}}{Median\ survival\ time_{Robotic}} \qquad (X)$$

$$SE\ Ln(HR) \approx \sqrt{\frac{1}{O_r} + \frac{1}{O_c}} \qquad (XI)$$

$$95\%\ CI\ Ln(HR) = Ln(HR) \pm 1.96 \times SE\ Ln\ (HR) \qquad (XII)$$

Where HR is hazard ratio, SE is standard error, "Ln" denotes the natural logarithm ($log_e$), $O_r$ and $O_c$ are the number of observed events for robotic and comparison groups, respectively, and CI is the confidence interval.

For papers where a time-to-event analysis was not performed and when we were unable to calculate a hazard ratio estimate using Method 1 through Method 4, dichotomous event data were summarized as a relative risk. In cases where zero event counts precluded estimation of a relative risk, a risk difference was used instead. The relative risks are not adjusted for time and censoring, which is problematic for the reasons described above and for this reason, we did not include them in the meta-analysis of HR.

## Data extraction methods

A data extraction form was created to capture all information related to time-dependent end-points, with separate tabs for overall survival, disease-free survival, and disease recurrence. This form had 6 categories of data; sample sizes (total number of patients in each cohort, number of patients included in each analysis, n at risk, and number of patients in matched or sub-group comparisons), Method 1 data (HR, CI low, CI high, p-value, statistical test, reference group and evidence for that designation), Method 2 data (the number of events or non-events in each cohort or the event proportions, the p-value and whether it was a log-rank, Cox proportional hazard, or other p-value, and the time point), Method 3 data (the figure # of the Kaplan-Meier curve), Method 4 data (the survival or death Kaplan-Meier estimate, the p-value and statistical test, and the time point of the estimate), and the conclusion of the authors for each comparison and each outcome (e.g., "the robotic cohort showed improved survival but no difference in disease-free survival compared to the open cohort"). A condensed version is shown in Table 2.

In addition, for each data type, the location in the paper where the data were found was documented to facilitate the quality control assessment. Statistical methods were also recorded, including any methods used to control for cohort imbalances (modeling with covariate adjustment, propensity score matching or weighting, other). Extractors were trained using a small subset of papers to improve the uniformity of data recording. Data were extracted as reported and all calculations were performed in a separate section of the spreadsheet to maintain an accurate accounting of all data, decisions, and calculations. Dual extraction was used to ensure data quality, and discrepancies were resolved by consensus with at least one additional reviewer [12].

## Quality control methods

Although the hierarchy was used to determine which hazard ratio to use for the analysis, when feasible, the estimates using the other methods were calculated for comparison. This strategy allowed for comparison of the magnitude and direction of the hazard ratio under various methods and helped to identify errors and other issues. Even when Method 3 was not used, if a Kaplan-Meier curve was presented and the hazards appeared relatively proportional, comparison of the hazard ratio to this graph at least for directionality was useful. Finally, comparison to the authors' conclusions was a valuable check to ensure that any assumptions made in Methods 1 through 4 did not lead to an inaccurate representation of the published data.

There were cases when the hazard ratios under multiple methods did not align. The most common reasons for discrepancies were related to comparing hazard ratios adjusted for covariates to estimates derived from unadjusted data, or in a few cases, distributions that would probably violate the proportional hazards assumption. We investigated differences across

**Table 2. Comparison of hazard ratios Method 1 through 4.** Comparison showing information required for calculation of hazard ratio using various methods and resulting hazard ratio and 95% confidence interval.

|  | N | Method 1 | Method 2 | Method 3 | Method 4a | Method 4b |
|---|---|---|---|---|---|---|
| Cohort |  | HR [95% CI] | Deaths | Est. event n | OS 3yr | Median Survival, Deaths |
| Robotic | 300 | Ref | 105 | 107 | 58.6% | 3.8 yr, 105 |
| Open | 300 | 1.47 [1.14, 1.90] | 133 | 132 | 44.5% | 2.5 yr, 133 |
| p-value |  | 0.0032 | 0.003 | 0.003 | 0.003 |  |
| Calculated HR [95% CI] |  | 0.68 [0.53, 0.88] | 0.68 [0.53, 0.88] | 0.68 [0.53, 0.88] | 0.71 [0.56, 0.89] | 0.66 [0.51, 0.85] |

methods until an explanation could be found and used the method that most aligned with our method hierarchy and the conclusions of the authors.

When more than one analysis was reported, we selected an adjusted or matched analysis preferentially, and only if no other data were available, unadjusted data; we chose the largest available analysis that adequately addressed cohort differences using the same hierarchy as listed above. When no adjustment or matching was performed, the comparability of the groups was determined by comparing baseline values for a list of covariates potentially related to oncologic outcomes. These two subgroups were analyzed and presented separately as well as combined for a total pooled result to identify any instances where adjustment provided a different result from analyses with comparable groups.

Due to the differences in the strength and number of assumptions needed for the various methods used, an analysis of the heterogeneity of outcomes based on method type should be performed.

## Results

### Worked example

We illustrate the four methods described above based on a simulated robotic versus open data set and we compare the resulting HR estimates (Table 2). For Method 1 the "reported" HR is: 1.47 [1.14, 1.9] with the robotic group as the reference. To switch to the open group as the reference, HR = 1/1.47 [1/1.9, 1/1.14] = 0.68 [0.53, 0.88]. Table 3 shows the worked example for Method 2, with the event n as reported in the paper entered in rows 7 (robotic) and 8 (open). The data for rows 7 and 8 can be obtained from one of three sources, directly from the manuscript (Method 2), estimated from the KM curve and Guyot algorithm (Method 3-Fig 3), and calculated from the KM survival estimate at the latest time point (Method 4a) by multiplying the survival estimate with n at risk for # alive, and then subtracting from n at risk to get the estimated number of patients who died. For the simulated data set, KM survival estimates at 3-years were 58.6% Robotic versus 44.5% Open, so the calculation would be 300-(58.6% x 300) = 124 for the robotic group and 300-(44.5% x 300) = 167 (Fig 3). For Method 4b, median survival and the event n can be used to calculate the HR and CI (Table 4). We are also providing the R code in S3 Appendix.

### Advantages of method hierarchy

To demonstrate the benefits of a hierarchal approach to hazard ratio extraction or estimation, the distribution of methods used in our analysis to date was calculated. Data from 199 papers were available for inclusion, with some reporting multiple outcomes. When limiting the analysis to papers that used adjustments or matching to account for differences between groups and papers where the groups were comparable, use of Methods 2–4 increased the available HRs from 108 (Method 1) to 240 HRs (Methods 1–4), facilitating an increase of 122%. Method 1 was the most used, accounting for about 45% of HRs across the various outcomes. About 15% of hazard ratios were derived using Method 2. Method 3 was the second most common with 28%, and Method 4 was the least common with 12%. The hierarchy of methods led to a dramatic increase in the number of papers that could be included in the analysis compared to restriction to publications reporting hazard ratios. To check the accuracy of including Methods 2–4, we performed simulations. In these simulations, Methods 2 and 4 resulted in hazard ratio estimates that fell within the original confidence interval >99% of the time. However, because it is difficult to simulate non-proportional hazards, truncated p-values, and other challenges in actual published reports, we examined all available publications for overall survival that provided a HR and allowed the use of at least one indirect method for estimating an HR.

**Table 3. Worked example using Method 2: Hazard ratio calculated using event counts.** HR = Hazard Ratio, CI = Confidence Interval, eq. = equation, est. = estimated, $V_r$ is the inverse variance of the log hazard ratio for the robotic group, $O_{Total}$ is the total number of events in the robotic plus the comparison group, "Ln" denotes the natural logarithm ($\log_e$), $\Phi^{-1}$ is the inverse of the standard normal distribution, and the p-value is assumed to be 2-sided and from the log-rank test if not otherwise stated. Because p-values were assumed to be two-sided, manual assignment of direction was adjusted by multiplying the $\Phi^{-1}\left(1-\frac{p}{2}\right)$ term by -1 or 1 so that the result was negative when survival in the robotic group was higher/better and positive when survival in the robotic group was lower/worse.

| | | D | F |
|---|---|---|---|
| | | Example with equations | Example with values |
| | Raw Data | Robotic vs Open | Robotic vs Open |
| 4 | $R_r$ (Total number of patients: robotic) | 300 | 300 |
| 5 | $R_c$ (Total number of patients: control) | 300 | 300 |
| 7 | $O_r$ (# deaths reported: robotic) | 105 | 105 |
| 8 | $O_c$ (# deaths reported: control) | 133 | 133 |
| 9 | Log-Rank p-value (KM or Cox PH) | 0.003 | 0.003 |
| | Calculations | | |
| 11 | Estimated death rate: robotic | = D7/D4 | 0.35 |
| 12 | Estimated death rate: control | = D8/D5 | 0.443 |
| 13 | Difference in est. death rate (r-c) | = D11-D12 | -0.093 |
| 14 | Direction of difference (enter: 1 if D13 is positive or -1 if D13 is negative) | -1 | -1 |
| 16 | eq. (V) = $V_r = (O_{total}*R_r*R_c)/(R_r+R_c)^2$ | = (((D7+D8)*D4*D5)/((D4+D5)^2)) | 59.5 |
| 17 | eq. (VI) = Variance Ln(HR) = $1/V_r$ | = 1/D16 | 0.0168 |
| 18 | eq. (VII) = $O_r - E_r = (\sqrt{(O_{total}*R_r*R_c)}/(R_r+R_c))*\Phi^{-1}(1\text{-p-value}/2)*(\text{direction of difference})$ | = (SQRT((D7+D8)*D4*D5)/(D4+D5))*(NORM.S.INV (1-D9/2)*D14) | -22.89 |
| 19 | eq. (VIII) = ln(HR) = $(O_r-E_r)/V_r$ | = D18/D16 | -0.385 |
| 21 | HR = $e^{Ln(HR)}$ | = EXP(D19) | 0.68 |
| 22 | 95% CI Lower = $e^{(Ln(HR)-1.96*\sqrt{(variance\ Ln(HR))})}$ | = EXP(D19-1.96*SQRT(D17)) | 0.53 |
| 23 | 95% CI Upper = $e^{(Ln(HR)+1.96*\sqrt{(variance\ Ln(HR))})}$ | = EXP(D19+1.96*SQRT(D17)) | 0.88 |

Comparing reported hazard ratios (Method 1) to estimated HR using Methods 2–4 within analysis type (ie. HR and KM both performed on matched cohort), showed that ≥90% of the estimated HRs fell within the 95% confidence interval of the reported HR (Method 2: 93%, Method 3: 96%, Method 4: 90%). The main reason papers fell outside of the confidence interval was that truncated p-values were reported (e.g. p<0.001). While this is not a fundamental inaccuracy of the method, it is an issue that arises in meta-analyses. Removing papers that report truncated p-values would result in a more precise, but less accurate analysis as it selectively removes papers with highly significant results. This type of check, along with sensitivity analyses based on publication quality, risk of bias, methods used, or number of additional

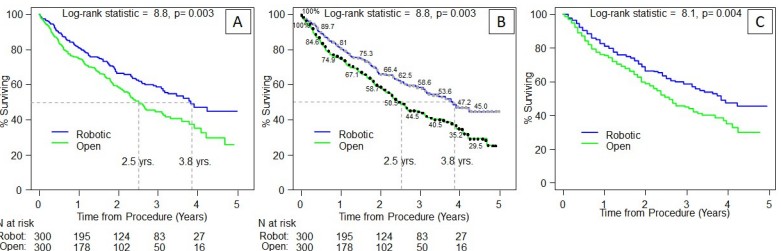

**Fig 3. Kaplan-Meier curve worked example: Example data for Method 3 Guyot algorithm and worked example.** Panel A is a graph that might appear in a publication. Panel B shows the "digitized" version with time and KM points, and panel C shows the re-constructed individual patient data using the digitization and n at risk as input. See S1 Appendix for full R code.

**Table 4. Worked example using Method 4b: Hazard ratio calculated using median survival estimates.**
HR = Hazard Ratio, CI = Confidence Interval, eq. = equation, "Ln" denotes the natural logarithm ($\log_e$).

| | | | D | F |
|---|---|---|---|---|
| | | | **Example with equations** | **Example with values** |
| | | **Raw Data** | **Robotic vs Open** | **Robotic vs Open** |
| 4 | | **Total number of patients: robotic** | 300 | 300 |
| 5 | | **Total number of patients: control** | 300 | 300 |
| 7 | | **$O_r$ (# deaths reported: robotic)** | 105 | 105 |
| 8 | | **$O_c$ (# deaths reported: control)** | 133 | 133 |
| 9 | | **$MS_r$ (Median Survival (in months): robotic)** | 3.8 | 3.8 |
| 10 | | **$MS_c$ (Median Survival (in months): control)** | 2.5 | 2.5 |
| | | **Calculations** | | |
| 12 | | *eq. (X)* = **HR = $MS_c/MS_r$** | = D10/D9 | 0.66 |
| 13 | | *eq. (XI)* = **Standard Error Ln(HR) = $\sqrt{(1/O_r+1/O_c)}$** | = SQRT(1/D7+1/D8) | 0.13 |
| 14 | | *eq. (XII)* = **CI Lower = ln(HR) − 1.96*SE Ln(HR)** | = LN(D12)-1.96*D13 | -0.67 |
| 15 | | *eq. (XII)* = **CI Upper = ln(HR) + 1.96*SE Ln(HR)** | = LN(D12)+1.96*D13 | -0.16 |
| 17 | | **Exponentiate CI Lower = $e^{(\ln(HR) - 1.96*SE\ Ln(HR))}$** | = EXP(D14) | 0.51 |
| 18 | | **Exponentiate CI Upper = $e^{(\ln(HR) + 1.96*SE\ Ln(HR)))}$** | = EXP(D15) | 0.85 |

assumptions required, can be useful to find the optimal balance between bias and precision based on the available literature.

## Discussion

This manuscript describes a hierarchical ordering of 4 methods for obtaining a hazard ratio or a hazard ratio estimate and illustrates and compares the results for a simulated example. It also provides recommendations for data extraction. Our methods add to the list of previously published techniques for analyzing time-to-event outcomes [4, 5, 8], and include details for implementing our strategies (S1, S3 and S4 Appendices—Guyot code, R functions for HR estimate calculations, and Data extraction & tricks). Finally, this report describes the motivation and assumptions underlying key process decisions.

The Cochrane Handbook recommends the methods we have labeled Methods 1–3 as valid ways to extract and pool hazard ratios for time-dependent data [7] and Tierney et al. 2020 [18], prioritized data extraction consistent with our ordering of Methods 1 through 3. They evaluated 18 systematic reviews in oncology with a preference for direct use of the hazards or hazard ratio, followed by estimation of the hazard ratio based on the survival analysis p-value and number of events, followed by extraction from the KM curve (though the extraction technique differed slightly [18]). However, neither of these prior publications provided the level of detail in performing these calculations demonstrated here. In practice, the preparation of data for meta-analysis is time consuming, requires many calculations (possibly with manual adjustments), and consists of a multitude of assumptions to check, caveats to consider, and ad hoc methods for special cases. This paper compiles multiple methods for calculating hazard ratios with clear instructions, a comprehensive worked example, practical tips, additional helpful calculations, and considerations relative to the assumptions. The goal was to provide enough information for others to reproduce this work.

The most important potential limitation is that the use of the hazard ratio to summarize time-to-event data may not be ideal. While certainly preferable to methods that do not account for time and censoring, the proportional hazards assumption is rarely evaluated in practice. An analysis of 115 trials found that only 4 described efforts to test the proportional hazards

assumption [19]. Interestingly, this study also demonstrated that the proportional hazards assumption is more likely to fail when treatments with different mechanisms of action are compared. There are reasonable alternatives to the hazard ratios that are easier to interpret when the assumption of proportional hazards is dubious. These include the Restricted Mean Survival Time, which is the average from time 0 to a specified point [19]. However, this estimate is seldom provided in practice, and its use would likely result in the exclusion of studies presenting only a hazard ratio.

Beyond evaluation in individual studies, the proportional hazards assumption is clinically questionable if the surgical approach only impacts short-term surgery related outcomes or the ability to fully excise the cancer. This assumption may be problematic when combining studies from vastly different follow-up durations, and an analysis restricted to longer studies (≥5 years) could be helpful to elucidate any issues when a sufficient number of longer studies become available. Extraction of IPD using the Guyot algorithm allows for investigation of the proportional hazards assumption and alternatives when invalid; we hope that the illustrations in 2 different statistical packages will make this powerful technique more accessible. However, the accuracy of this method depends on the quality of Kaplan-Meier figures and the completeness of the manuscript in reporting n at risk. An analysis of 125 oncology publications found that any post-baseline n at risk information was reported for about half of the manuscripts, and n at risk for at least 4 time-points was reported for about a third [20]. Our findings were similar. More journals are starting to require this information, but until this requirement becomes ubiquitous, IPD extraction will be a useful alternative but not a panacea.

When the hazard ratio is a reasonable choice, the next consideration is the extraction process. The rationale for developing our process was to maximize the number of papers included in the analysis, while consistently providing the best possible estimate of the hazard ratio based on the data reported. These simultaneous goals necessitate the inclusion of lower quality publications (from a level of evidence/risk of bias standpoint) in the analysis. In addition, the assumptions required to include some papers cannot be assessed using the published results (e.g., the assumption of proportional hazards when no Kaplan-Meier curve was available). Such assumptions, if invalid, could potentially decrease the accuracy of the final pooled result. However, the alternative of limiting meta-analyses of time-dependent outcomes to publications reporting hazard ratios can greatly reduce the number of papers available for analysis, alter the conclusions, and introduce bias. The addition of methods 2–4 allowed us to include manuscripts that did not provide a hazard ratio, and the decision tree allowed us to be systematic in choosing the best possible estimation method when data were available for more than one method. Overall, long-term outcomes were reported in 199 articles in some form. Use of Methods 2–4 increased the available HRs from 108 (Method 1) to 240 HRs (Methods 1–4), facilitating an increase of 122% in the included outcome assessments. The meta-analysis technique assumes that all relevant studies are available (fixed model), or at least a random sample of studies (random model) can be used. Failure to include studies that do not present hazard ratios creates a special case of reporting bias; it is possible that the reporting of a hazard ratio correlates with other factors, such as the involvement of a statistician, degree of a priori analysis specification, affluence, centers of excellence, etc. that could cause systemic bias and alter the conclusions. More stringent selection criteria including analysis methods could reduce the generalizability of the conclusions by limiting surgeon experience and patient characteristics.

Our decision to maximize studies and patients also mandated inclusion of a wider range of publication types, including retrospective comparative studies. In general, the appropriateness of this decision depends on the clinical context of the project. The inclusion of observational data in meta-analyses is controversial as the heterogeneity and risk of bias within and across studies is much higher than for randomized trials [21]. Concerns about individual and

aggregate observational study results are warranted. In our project, some observational studies used techniques such as Cox proportional hazards regression and propensity scores to reduce selection bias and confounding. However, differences in analysis methods and adjustment factors may increase heterogeneity or raise questions about comparability [21]. In our case, the inclusion of observational studies was imperative; of the 199 total manuscripts with time-dependent outcomes identified, only 7 of them were randomized trials and two of the surgical procedures had no RCT representation at all. There are very few randomized controlled trials that have long enough follow up to be able to report on survival outcomes.

In light of the paucity of RCTs, the question is how to have confidence in a result based primarily on observational data. We are assuming that the benefits of increasing the number of publications outweigh the additional risk of bias that may arise from including non-randomized comparison papers reporting long-term outcomes of interest. Formal assessment of risk of bias is standard in reporting results, and tools are available for both randomized (e.g. Cochrane) and non-randomized (e.g. Robins-I, Ottawa Newcastle) studies. These, or other measures of bias risk along with sensitivity analyses, could be used to further investigate the trade-off between including a broader cross section of the literature and the increased risk of bias.

Though less reliable than randomized trials, there are some advantages to observational studies. Randomized trials are prone to spectrum bias when onerous inclusion and exclusion criteria eliminate large swaths of the patient population, and a protocol-driven, standardized approach to clinical decision-making may differ markedly from real-world practice [22]. Novel surgical approaches are often adopted by a few innovative centers and are not well-represented in national databases (SEER, NCDB) for many years after introduction. Randomization remains the best technique for the removal of systemic imbalances, and whether observational data can produce results with similar validity will continue to be hotly contested (as noted by a recent publication titled, "The magic of randomization versus the myth of real-world evidence" [23]). However, limiting our analysis to papers that performed adjustment or matching or to papers where the cohorts were comparable helps mitigate these issues.

## Conclusions

Meta-analysis of time-dependent outcomes using hazard ratios can produce valid synthesis, and the ability to obtain hazard ratio estimates from the available information can aid in this goal. Though meta-analyses pooling hazard ratios are easier to interpret when the proportional hazards assumption holds, the hazard ratio is a valid summary statistic, adjusted for time and censoring, even when this assumption is violated. Providing a practical guide for the implementation of best practices for long-term outcome analysis based on hazard ratios will hopefully reduce methodological obstacles that currently preclude them. Sensitivity analyses can be added as needed to address issues such as heterogeneity across results. Future studies or technical advancement proposing methodology to streamline the implementation of the Guyot method and software automating the calculations based on the formulae described above would also be useful.

## Supporting information

**S1 Appendix. Guyot SAS and R-code and examples.**
(DOCX)

**S2 Appendix. Validation of Guyot algorithm.**
(DOCX)

**S3 Appendix. R functions for HR estimate calculations.**
(DOCX)

**S4 Appendix. Assumptions, rules, and tips.**
(DOCX)

## Author Contributions

**Conceptualization:** April E. Hebert, Usha S. Kreaden, Angela B. Soito, April E. Slee.

**Data curation:** April E. Hebert, Ana Yankovsky, Dongjing Guo, Samira Massachi, April E. Slee.

**Formal analysis:** April E. Hebert, Yang Li, Shih-Hao Lee, Yuki Liu, April E. Slee.

**Funding acquisition:** Usha S. Kreaden.

**Methodology:** Usha S. Kreaden, Angela B. Soito, April E. Slee.

**Project administration:** Usha S. Kreaden, Angela B. Soito.

**Software:** April E. Hebert, Yang Li, Yuki Liu, April E. Slee.

**Supervision:** Usha S. Kreaden, Angela B. Soito.

**Validation:** Yang Li, Shih-Hao Lee, Yuki Liu, April E. Slee.

**Visualization:** April E. Hebert, April E. Slee.

**Writing – original draft:** April E. Hebert, Usha S. Kreaden, Dongjing Guo, Yang Li, Angela B. Soito, April E. Slee.

**Writing – review & editing:** Shih-Hao Lee.

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
