## [Decision Letter · Decision Letter 0]

4 Aug 2021

PONE-D-21-18912

Methodology to Standardize Heterogeneous Statistical Data Presentations for Combining Time-to-Event Oncologic Outcomes

PLOS ONE

Dear Dr. Slee,

Thank you for submitting your manuscript to PLOS ONE. After careful consideration, we feel that it has merit but does not fully meet PLOS ONE’s publication criteria as it currently stands. Therefore, we invite you to submit a revised version of the manuscript that addresses the points raised during the review process.

We look forward to receiving your revised manuscript.

Kind regards,

Mona Pathak, PhD

Academic Editor

PLOS ONE

Journal Requirements:

Reviewers' comments:

Reviewer's Responses to Questions

**Comments to the Author**

1. Is the manuscript technically sound, and do the data support the conclusions?

Reviewer #1: Partly

Reviewer #2: Yes

2. Has the statistical analysis been performed appropriately and rigorously? 

Reviewer #1: No

Reviewer #2: Yes

3. Have the authors made all data underlying the findings in their manuscript fully available?

Reviewer #1: No

Reviewer #2: Yes

4. Is the manuscript presented in an intelligible fashion and written in standard English?

Reviewer #1: Yes

Reviewer #2: Yes

5. Review Comments to the Author

Reviewer #1: This interesting manuscript outlines a decision process for selecting effect sizes, and associated precision, from articles (and other reports) of survival (overall, disease-free, and recurrence) where these are not explicitly presented using hazard ratios and CIs/SEs, covering both observational and RCT study designs. In a case study, they demonstrate that many more results (and more studies) could be included in a meta-analysis compared to limiting inclusion to those reporting HRs and CIs or SEs.

While increasing the number of results able to be included is admirable, as a biostatistician, I would also have significant concerns about the bias versus precision trade-off inherent in this hierarchy. While increasing from 115 HRs to 303, all other things being equal, would be a highly desirable outcome, the additional noise and possible biases from the approximations seems worth more attention here, and not just study quality considerations. Have you performed simulations to identify the magnitude of numerical bias introduced by using these approximations? Can you summarise results from simulation-based articles if not? I appreciated the consideration of this point in the discussion, I’m wondering here about practical advice to the reader about how to approach this trade-off.

I was surprised to see no mention of competing risks survival analyses, as far as I could tell, particularly given that I would expect prognostic questions to be much more important here. For example, both disease-free and recurrence outcomes would have other-cause mortality as a competing risk for prognostic research questions. I was expecting that the presence of a competing risk event would be explicitly mentioned at the end of the second paragraph on the second page, along with “patient withdrawal and loss to follow-up.” This also presents a general challenge in systematic reviews and meta-analyses in oncology, and many other areas, where both HRs and SHRs are likely to be reported. I wonder if the authors could incorporate competing risks at appropriate places into their manuscript.

As a minor comment, I wonder if explaining the use of Phi in equation I (and subsequent) might not be a kindness to the non-statistical reader (this information is eventually revealed after Equation III), perhaps along with its value for (at least) 95% CIs. Similarly, some readers might appreciate being reminded about alpha here and subsequently. I think readers should be very clear about all notation immediately after reading any equation, even when the notation is as standard as it is here.

As another minor comment, “univariate” and “multivariate” are generally used by (bio)statisticians to refer to dependent variables, with “univariable” and “multivariable” (referring to independent variables) perhaps being the intention on the second paragraph after Equation II (and elsewhere)? More importantly, in this paragraph, I wonder if readers would benefit from some clearer explanation earlier in the manuscript of what to do when “multiple hazard ratios were reported” in terms of different sets of adjustment variables or when complete case and imputation results are presented, as examples? While RCTs ought not to include confounders, unless there are differential missing data mechanisms, favouring models that include variables used in allocation (stratification or minimisation variables) and competing exposures, while excluding variables potentially on the causal pathway, might be useful advice (perhaps brief with references to further information) here? The advice in “Specifically, the order of preference in this analysis for HRs was: unmatched unadjusted or univariate, unmatched adjusted, unmatched multivariate, matched unadjusted or univariate, matched adjusted, matched multivariate.” and in Figure 1 to favour unadjusted comparisons and those that break matching both go against standard (bio)statistical advice in my view (as a biostatistician) and I think need very careful and well-referenced justification in the body of the manuscript here. “Downgrading” all reported associations to the lowest common denominator, which seems to be the goal here I think, does not strike me as the obvious preference and needs some strong justification in my view. I appreciate that more discussion of this point is in the supplement (S3), but I think this needs to be incorporated into the manuscript, even if only briefly.

Method 4 seems the most open to challenge, also not being a Cochrane “standard” approach, and as the authors note this make assumptions including there being no censoring (or at least censoring mechanisms that are equivalent across treatments). While I appreciate their comment about including this as a limitation, I wondered if a sensitivity analysis with such studies being removed might not also be an appropriate treatment here. Why is using median survival not explicitly included in Figure 1 given its discussion in the text (this involves two time points rather than one and so seems distinct to me)?

I think all four methods would be more likely to be understood by readers, and so used appropriately, if “worked” examples of each were added to the body of the manuscript. Methods I and II should be simple enough here, method III would be nicely illustrated with a figure, and method IV could be usefully shown using both survival through to a fixed timepoint and median survival. I do appreciate the references the authors have included, but I think that the greatest value of this manuscript would be as a guide and a tutorial at the same time, and so, on that basis, I feel that it should be reasonably self-contained (as much as is possible). I appreciate that the supplement (S3) includes much important information about the practice of using these methods, but even there, I’d like to see the essentials moved into the manuscript as I fear many readers will not read the supplements in detail (or will not connect some of the “tips” to the process explained within the body of the manuscript). Your rule to use the most conservative HR, for example, might be useful enough to warrant being included in the manuscript. Similarly, dealing with p-values reported as inequalities, seems rather crucial to me.

I don’t wish to be snobbish, but the 3D exploded pie chart in Figure 4 seems to me to be a very questionable way of presenting these four values (the information to ink ratio is very low here) and could easily be deleted given that this information is presented in the text.

As a final minor comment, I suggest not using “subjects”, as in “when the goal was to maximize the number of included papers and the number of included subjects” (but note at least four other such uses) and instead using “patients” or (if you feel it is appropriate) “participants”.

Reviewer #2: Highly appreciated the author’s effort to consolidate methods for calculating hazard ratio, the most appropriate statistic for a meta-analysis of time dependent outcomes, with clear instructions, a comprehensive worked example, practical tips, and additional calculations. Please address the following concerns

Query 1: Pages 10 to 12: Use of roman numerals to quote equation number prior to the equation. It may be kept after the equation in the right margin for further use

Query 2: P14, Line 6 : While describing Method 3, the authors specified that “When the

overall p-value seemed to reflect pairwise differences (assumption of similar variances), then the 3-way p-value was assumed to be a reasonable approximation of each pairwise comparison”. This may increase type 1 error rate due to multiple pairwise comparison. How do you address the difference in type I error rate due to each pairwise comparison??

Query 3: Decision Tree for Hazard Ratio Extraction specified in Figure 1 is remarkable. However the method 4 mentioned by the authors hold the assumption of no censoring; which is difficult to consider in real research studies. How do you justify this?

6. PLOS authors have the option to publish the peer review history of their article (what does this mean?). If published, this will include your full peer review and any attached files.

Reviewer #1: **Yes: **Andrew R Gray

Reviewer #2: No

---

## [Author Response · Author response to Decision Letter 0]

29 Oct 2021

We wish to re-submit the original research article (PONE-D-21-18912) entitled: “Methodology to Standardize Heterogeneous Statistical Data Presentations for Combining Time-to-Event Oncologic Outcomes” for consideration by PLOS ONE. 

We have addressed the reviewers’ comments and we have uploaded a rebuttal letter that responds to each point raised by the academic editor and reviewer(s) in a file: ‘Response to Reviewers’. This file also outlines the changes to the manuscript, as requested. We are also uploading a tracked changes version of the manuscript ('Revised Manuscript with Track Changes modified’) and a clean version of the manuscript (‘Manuscript modified’) with all of the proposed changes.

We also updated the Financial Statement to reflect that all authors are employees or consultants of Intuitive Surgical and all work related to this paper was performed in the course of usual work operations and was not based on a grant or any other study-specific funding award as follows:

“Funding Statement

All authors are employees of Intuitive Surgical, Inc. (AEH, USK, AY, DG, YLi, SL, YLiu) or consult for Intuitive Surgical, Inc. (ABS, SM, AES) and all work related to this paper was performed in the course of usual work operations and was not based on a grant or other study-specific funding award.”

We thank the reviewers for their time and their thoughtful suggestions, which we think strengthen the manuscript.

PONE-D-21-18912

Methodology to Standardize Heterogeneous Statistical Data Presentations for Combining Time-to-Event Oncologic Outcomes

PLOS ONE

Response to Reviewers’ Comments:

Reviewer #1: This interesting manuscript outlines a decision process for selecting effect sizes, and associated precision, from articles (and other reports) of survival (overall, disease-free, and recurrence) where these are not explicitly presented using hazard ratios and CIs/SEs, covering both observational and RCT study designs. In a case study, they demonstrate that many more results (and more studies) could be included in a meta-analysis compared to limiting inclusion to those reporting HRs and CIs or SEs.

Question #1:

While increasing the number of results able to be included is admirable, as a biostatistician, I would also have significant concerns about the bias versus precision trade-off inherent in this hierarchy. While increasing from 115 HRs to 303, all other things being equal, would be a highly desirable outcome, the additional noise and possible biases from the approximations seems worth more attention here, and not just study quality considerations. Have you performed simulations to identify the magnitude of numerical bias introduced by using these approximations? Can you summarise results from simulation-based articles if not? I appreciated the consideration of this point in the discussion, I’m wondering here about practical advice to the reader about how to approach this trade-off.

Response to Question #1:

There are two issues at play: whether these methods work in theory (which can be addressed through simulation studies) and whether they work in practice (with real-world data and reporting issues). To address whether these work in theory, we have performed simulations. We had already run simulations testing Method 3 (see Supplemental Appendix S2) and we utilized the same data set to run simulations for Methods 2 and 4. 

We found extremely high accuracy for all methods. For Method 3, the results of these simulations are reported in Supplemental Appendix S2 and they show that if the proportion censored is less than 95% and n at risk are provided, the estimated HR is very accurate. This finding matches the simulation studies conducted by Guyot {Guyot P, Ades AE, Ouwens MJNM, Welton NJ. Enhanced secondary analysis of survival data: reconstructing the data from published Kaplan-Meier survival curves. BMC MED RES METHODOL. 2012;12:9}. For Method 2, 1300 simulations resulted in a mean of 0.000 SD 0.017 and showed an HR ratio of 1.000 [0.999, 1.001] and for Method 4, 1300 simulations resulted in a mean of 0.001 SD 0.019 and showed an HR ratio of 1.001 [0.999, 1.002]. This is consistent with the asymptotic properties of these estimators {Kalbfeisch, J. D. and Prentice, R. L. The Statistical Analysis of Failure time Data, Oxford University Press, Oxford, 1980; Tsiatis, A. A. The asymptotic joint distribution of the efficient scores test for the proportional hazards model calculated over time. Biometrika, 68(1), 311-315 (1981)}.

Unfortunately, there are many factors that impact the accuracy of these estimates that are difficult to simulate. Some are issues of reporting, such as truncated curves, truncated p-values, and missing information. Others are related to the shape of the Kaplan-Meier curves themselves (deviations from proportional hazards). Therefore, to address whether these work in practice, we performed an accuracy check of the HRs estimated using Methods 2-4 compared to the HRs reported by the authors for one outcome. Papers that reported an HR and sufficient data to estimate at least one additional HR using Methods 2-4 were selected. Analyses based on similar data (ie. matched Method 1 HR vs. matched Method 2 HR) were compared by determining the number of estimated HRs that fell within the 95% confidence interval of the reported HR. Note that this is not an entirely unbiased assessment of error because most manuscripts that presented proportional hazards models did so because the authors were suspicious about residual confounding.

(please see figure provided in 'response to reviews' file)

Even in these suboptimal conditions (because the presence of an HR implies the need to adjust for confounders), ≥ 90% of the estimated HRs fell within the 95% confidence interval of the reported HR (Method 2: 93%, Method 3: 96%, Method 4: 90%). The main reason papers fell outside of the confidence interval was because truncated p-values were reported (e.g. p<0.001). While this is not a fundamental inaccuracy of the method, it is an issue that arises in meta-analyses. Removing papers that report truncated p-values would result in a more precise, but less accurate analysis as it selectively removes papers with highly significant results. 

In addition, a larger study assessed the accuracy of the 4 most popular methods for extracting data from Kaplan-Meier curves (Guyot, Parmar, Williamson, Hoyle and Henley) (our Method 3) using a similar technique and including nearly 200 published Kaplan-Meier curve/hazard ratio pairs from regulatory submissions of cancer treatments. This study found that the Guyot technique produced the most accurate estimates and the smallest bias of the four techniques. The error rate on the hazard scale was immaterial (HR: 1.0094 and 95% CI 0.998 to1.020) {Saluja R, Cheng S, Delos Santos KA, Chan KKW. Estimating hazard ratios from published Kaplan-Meier survival curves: A methods validation study. Res Synth Methods. 2019 Sep;10(3):465-475. doi: 10.1002/jrsm.1362. Epub 2019 Jun 24. PMID: 31134735}.

We also agree that it is important to make the reader aware of these issues. We have added a recommendation to perform sensitivity analyses based on method and risk of bias or quality assessments and have made the following changes to the manuscript as follows:

Methods, p-value section: “When using a p-value to calculate an HR or CI, if it is reported as “less than” (e.g., p<0.001), check for a log-rank statistic and, if reported, use it to derive a more precise p-value. If no log-rank statistic is reported, the convention is to treat it as equal (e.g., p<0.001 becomes p=0.001). This convention has the effect of biasing the resulting estimates toward the null hypothesis.”

Method 4 methods: “We strongly recommend a sensitivity analysis excluding Method 4 to understand the possible impact of these assumptions on the overall conclusions.”

Advantages of Method Hierarchy: “To check the accuracy of including Methods 2-4, we performed simulations. In these simulations, Methods 2 and 4 resulted in hazard ratio estimates that fell within the original confidence interval >99% of the time. However, because it is difficult to simulate non-proportional hazards, truncated p-values, and other challenges in actual published reports, we examined all available publications for overall survival that provided a HR and allowed the use of at least one indirect method for estimating an HR. Comparing reported hazard ratios (Method 1) to estimated HR using Methods 2-4 within analysis type (ie. HR and KM both performed on matched cohort), showed that ≥90% of the estimated HRs fell within the 95% confidence interval of the reported HR (Method 2: 93%, Method 3: 96%, Method 4: 90%). The main reason papers fell outside of the confidence interval was that truncated p-values were reported (e.g. p<0.001). While this is not a fundamental inaccuracy of the method, it is an issue that arises in meta-analyses. Removing papers that report truncated p-values would result in a more precise, but less accurate analysis as it selectively removes papers with highly significant results. This type of check, along with sensitivity analyses based on publication quality, risk of bias, methods used, or number of additional assumptions required, can be useful to find the optimal balance between bias and precision based on the available literature.”

Question #2

I was surprised to see no mention of competing risks survival analyses, as far as I could tell, particularly given that I would expect prognostic questions to be much more important here. For example, both disease-free and recurrence outcomes would have other-cause mortality as a competing risk for prognostic research questions. I was expecting that the presence of a competing risk event would be explicitly mentioned at the end of the second paragraph on the second page, along with “patient withdrawal and loss to follow-up.” This also presents a general challenge in systematic reviews and meta-analyses in oncology, and many other areas, where both HRs and SHRs are likely to be reported. I wonder if the authors could incorporate competing risks at appropriate places into their manuscript.

Response to Question #2:

This is a good point. To address this, we have added the following statement to the manuscript as marked with underlining:

Introduction: “Other common reasons that an event was not observed during a study include patient withdrawal, a competing risk event (an event that precludes observing the outcome of interest), and loss to follow-up.”

Question #3

As a minor comment, I wonder if explaining the use of Phi in equation I (and subsequent) might not be a kindness to the non-statistical reader (this information is eventually revealed after Equation III), perhaps along with its value for (at least) 95% CIs. Similarly, some readers might appreciate being reminded about alpha here and subsequently. I think readers should be very clear about all notation immediately after reading any equation, even when the notation is as standard as it is here.

Response to Question #3:

Thank you for the suggestion. To address this, we have added details to the description of each equation so that all symbols are identified and defined.

Question #4

As another minor comment, “univariate” and “multivariate” are generally used by (bio)statisticians to refer to dependent variables, with “univariable” and “multivariable” (referring to independent variables) perhaps being the intention on the second paragraph after Equation II (and elsewhere)? 

Response to Question #4: 

The preferred wording has been added in place of “univariate” and “multivariate” in all places where they are used in the manuscript.

Question #5

More importantly, in this paragraph, I wonder if readers would benefit from some clearer explanation earlier in the manuscript of what to do when “multiple hazard ratios were reported” in terms of different sets of adjustment variables or when complete case and imputation results are presented, as examples? While RCTs ought not to include confounders, unless there are differential missing data mechanisms, favouring models that include variables used in allocation (stratification or minimisation variables) and competing exposures, while excluding variables potentially on the causal pathway, might be useful advice (perhaps brief with references to further information) here? The advice in “Specifically, the order of preference in this analysis for HRs was: unmatched unadjusted or univariate, unmatched adjusted, unmatched multivariate, matched unadjusted or univariate, matched adjusted, matched multivariate.” and in Figure 1 to favour unadjusted comparisons and those that break matching both go against standard (bio)statistical advice in my view (as a biostatistician) and I think need very careful and well-referenced justification in the body of the manuscript here. “Downgrading” all reported associations to the lowest common denominator, which seems to be the goal here I think, does not strike me as the obvious preference and needs some strong justification in my view. I appreciate that more discussion of this point is in the supplement (S3), but I think this needs to be incorporated into the manuscript, even if only briefly.

Response to Question #5:

After further consideration, we agree with the reviewer. We have made changes throughout the manuscript to reflect that papers and analyses that have adequately controlled confounding should be prioritized. All numbers and % were updated to reflect the change to limiting the analysis to adjusted/balanced data only. The changes to the manuscript are as follows:

Abstract: “In our example, use of the proposed methodology would allow for the increase in data inclusion from 108 hazard ratios reported to 240 hazard ratios reported or estimated, resulting in an increase of 122%.”

Introduction: “The goal in our framework development was to maximize the number of included studies while limiting bias, to provide clear guidelines, and improve agreement in dual data extraction of each individual manuscript.”

Method 1 methods: “It is important to determine an extraction preference a priori for when more than one hazard ratio is reported. Our criterion was to maximize group size because analyses using entire populations account for the relative frequency of case types, severity of disease, surgeon experience, etc. and the results are more generalizable. For these reasons, we prioritized an adjusted HR using the largest sample that adequately addresses confounding (ie. whole patient population over a matched patient cohort when matching decreased the sample size).” 

Quality Control Methods: “When more than one analysis was reported, we selected the largest available analysis that adequately addressed cohort differences using the same hierarchy as listed above. When no adjustment or matching was performed, the comparability of the groups was determined by comparing baseline values for a list of covariates potentially related to oncologic outcomes. These two subgroups were analysed and presented separately as well as combined for a total pooled result to identify any instances where adjustment provided a different result from analyses with comparable groups.”

Quality Control Methods: deleted the following section: 

“A comparison of the results restricted to the “matched/balanced” studies to the overall results allowed us to assess the impact of selection bias and other kinds of confounding. Alternatively, when the goal was to maximize the number of included papers and the number of included patients, the selection differed slightly. For example, when hazard ratios were not presented, the available information was usually based on the unadjusted empirical estimates of the cumulative distribution function (Kaplan-Meier (KM) estimates). This information is most similar to univariable proportional hazards models with no covariate adjustment, so when multiple HR estimates were presented, the univariable ones were used. Specifically, the order of preference in this analysis for HRs was: unmatched unadjusted or univariable, unmatched adjusted, unmatched multivariable, matched unadjusted or univariable, matched adjusted, matched multivariable.” 

Advantages of Method Hierarchy: “When limiting the analysis to papers that used adjustments or matching to account for differences between groups and papers where the groups were comparable, use of Methods 2-4 increased the available HRs from 108 (Method 1) to 240 HRs (Methods 1-4), facilitating an increase of 122%. Method 1 was the most commonly used, accounting for about 45% of HRs across the various outcomes. About 15% of hazard ratios were derived using Method 2. Method 3 was the second most common with 28%, and Method 4 was the least common with 12%.”

Discussion: “In light of the paucity of RCTs, the question is how to have confidence in a result based primarily on observational data. We are assuming that the benefits of increasing the number of publications outweigh the additional risk of bias that may arise from including non-randomized comparison papers reporting long-term outcomes of interest. Formal assessment of risk of bias is standard in reporting results, and tools are available for both randomized (e.g. Cochrane) and non-randomized (e.g. Robins-I, Ottawa Newcastle) studies. These, or other measures of bias risk along with sensitivity analyses, could be used to further investigate the trade-off between including a broader cross section of the literature and the increased risk of bias.”

Discussion: “Randomization remains the best technique for the removal of systemic imbalances, and whether observational data can produce results with similar validity will continue to be hotly contested (as noted by a recent publication titled, “The magic of randomization versus the myth of real-world evidence”(24)). However, limiting our analysis to papers that performed adjustment or matching or to papers where the cohorts were comparable helps mitigate these issues.”

Figure 1: changed order of HR preference in the decision tree to: adjusted, then matched, then unadjusted/unmatched.

Question #6:

Method 4 seems the most open to challenge, also not being a Cochrane “standard” approach, and as the authors note this make assumptions including there being no censoring (or at least censoring mechanisms that are equivalent across treatments). While I appreciate their comment about including this as a limitation, I wondered if a sensitivity analysis with such studies being removed might not also be an appropriate treatment here. Why is using median survival not explicitly included in Figure 1 given its discussion in the text (this involves two time points rather than one and so seems distinct to me)?

Response to Question #6:

We agree that Method 4 is the most open to challenge and we agree that a sensitivity analysis excluding this method could be important, especially if strong statements about the pooled results will be made. Therefore, we have added the following statement to the Method 4 Methods section of the manuscript as follows:

“We strongly recommend a sensitivity analysis excluding Method 4 to understand the possible impact of these assumptions on the overall conclusions.”

We also performed an accuracy check for the indirect methods, including method 4, as mentioned above in our response to question #1

There were very few instances where median survival was the only data provided in our literature, and even though there are two different time points, there is still only one time point per cohort and this method also assumes no censoring, so we felt that it was appropriate to group it with Method 4. 

We added this to the manuscript as follows: 

“Method 4 was used in instances when a time-dependent analysis was performed, but it was not possible to account for the censoring distribution because the reported information was limited to a single time point per cohort, either in the form of Kaplan-Meier estimates, or median survival.”

We are aware that median survival is more commonly reported in other clinical settings, so we have changed it to Method 4b and added a separate mention to Figure 1 to read: 

“Are number of events, Kaplan-Meier estimates for median survival available?” ->Method 4B: Assume no censoring, constant event rate. Use median survival in each group to estimate HR.”

We have also added the equations for calculating the HR and confidence interval from median survival to the manuscript as follows:

(please see new equations in 'response to reviewers' file)

Where HR is hazard ratio, SE is standard error, “Ln” denotes the natural logarithm (loge), Or and Oc are the number of observed events for robotic and comparison groups, respectively, and CI is confidence interval.”

Question #7

I think all four methods would be more likely to be understood by readers, and so used appropriately, if “worked” examples of each were added to the body of the manuscript. Methods I and II should be simple enough here, method III would be nicely illustrated with a figure, and method IV could be usefully shown using both survival through to a fixed timepoint and median survival. I do appreciate the references the authors have included, but I think that the greatest value of this manuscript would be as a guide and a tutorial at the same time, and so, on that basis, I feel that it should be reasonably self-contained (as much as is possible). I appreciate that the supplement (S3) includes much important information about the practice of using these methods, but even there, I’d like to see the essentials moved into the manuscript as I fear many readers will not read the supplements in detail (or will not connect some of the “tips” to the process explained within the body of the manuscript). Your rule to use the most conservative HR, for example, might be useful enough to warrant being included in the manuscript. Similarly, dealing with p-values reported as inequalities, seems rather crucial to me.

Response to Question #7:

We have made sure that all of the assumptions and tips listed in Supplemental Appendix S3 have been added to the main body of the manuscript, except for those specific to biochemical recurrence in prostate cancer as follows (additions marked with underlining): 

Methods to Extract Hazard Ratios: “The individual Methods 1-4 are described in detail below. Methods 1 through 3 have previously been described(4, 7, 10) and are recommended by the Cochrane Handbook(8). Our base assumption was that hazard ratios are a valid comparison of overall risk between groups in directionality and magnitude even when the hazards are not proportional, but statements quantifying the comparisons (e.g., a 5 x higher risk) should not be made in the case of non-proportionality. Our main rules were that 1) all available data, outcome definitions, and stated conclusions were utilized to determine the most valid data, method, and p-value to use, and to check the accuracy of Method 2-4 calculations, 2) when there was a judgement call needed, we selected the method that was the most conservative (most disfavored) for the cohort of interest.”

Method 2: “For overall survival, the total number of deaths may be calculated by summing across causes of death (e.g., dead of other (DOO) + dead of disease (DOD)), subtracting the number of alive from the total sample size, or by calculating from a proportional death rate. For composite endpoints such as disease-free survival, it is important to be cautious to avoid double-counting patients that experienced multiple events (a patient that experiences recurrence and death would be counted twice if recurrence event n and a death event n were summed), but if there is no evidence it would be inaccurate, DOO + DOD + alive with disease (AWD) or overall mortality + (total recurrence minus DOD) equations could be used.”

Method 3: “The use of Method 3 is associated with additional considerations. If the number of events is few enough, manually counting them may be an option, but the result should be confirmed by manually calculating the KM estimate. However, the quality of the published image affects the ability to accurately count or digitize the Kaplan-Meier curve.”

“If no KM curve is shown, but the time of each event was reported along with summary information about the follow-up distribution, an approximation to the KM curve can be constructed manually.”

“The Log-Rank p-value can be used to adjust censoring in the Guyot algorithm when n at risk over time is not reported.”

Method 4: “We determined a priori which timepoint would be used when several Kaplan-Meier estimates were provided. We preferentially used the latest timepoint. Method 2 was then applied to these estimated event counts. This approach assumes no censoring, that the survival curves do not cross after the estimated time point, and that the hazards are relatively proportional…“

“We also utilized the conclusions of the authors to determine if this approach would accurately reflect the overall comparison between cohorts, and discrepancies were cause for excluding the data if the results and conclusions conflicted and the correct result was unclear.”

“Using the median survival produces a reasonable estimate when the cohort sizes are similar and when there is a constant event rate.”

Note that the key difference between Method 2 and Methods 3 and 4a is that in Method 2, the number of events is explicitly reported, while in Methods 3 and 4a, the number of events is estimated. Thus, we have added a table showing the calculation for Method 2 in full detail and we have added a description in a figure of how to estimate the number of events from a Kaplan-Meier curve using the Guyot algorithm. We have also added illustrated in text how to estimate the number of events using the Kaplan-Meier probability, and these estimated event numbers can be plugged into the Method 2 calculator to obtain Method 3 and 4a HR estimates. We have also added a table illustrating how to derive an HR estimate from the median survival times. We hope that these additions will satisfy the request for “worked” examples.

We have added the above content in an additional section in the manuscript titled: “Worked Example” with corresponding tables and figures as follows: 

“Worked Example:

We illustrate the four methods described above based on a simulated robotic versus open data set and we compare the resulting HR estimates (Table 2). For Method 1 the “reported” HR is: 1.47 [1.14, 1.9] with the robotic group as the reference. To switch to the open group as the reference, HR=1/1.47 [1/1.9, 1/1.14] = 0.68 [0.53, 0.88]. Table 3 shows the worked example for Method 2, with the event n as reported in the paper entered in rows 7 (robotic) and 8 (open). The data for rows 7+8 can be obtained from one of three sources, directly from the manuscript (Method 2), estimated from the KM curve and Guyot algorithm (Method 3-Figure 4), and calculated from the KM survival estimate at the latest time point (Method 4a) by multiplying the survival estimate with n at risk for # alive, and then subtracting from n at risk to get the estimated number of patients who died. For the simulated data set, KM survival estimates at 3-years were 58.6% Robotic versus 44.5% Open, so the calculation would be 300-(58.6% x 300)=124 for the robotic group and 300-(44.5% x 300)=167 (Figure 3). For Method 4b, median survival and the event n can be used to calculate the HR and CI (Table 4).”

 

Table 3: Worked Example using Method 2: Hazard ratio calculated using event counts

 D F

 Example with equations Example with values

 Raw Data Robotic vs Open Robotic vs Open

4 Rr (Total number of patients: robotic) 300 300

5 Rc (Total number of patients: control) 300 300

7 Or (# deaths reported: robotic) 105 105

8 Oc (# deaths reported: control) 133 133

9 Log-Rank p-value (KM or Cox PH) 0.003 0.003

 Calculations 

11 Estimated death rate: robotic =D7/D4 0.35

12 Estimated death rate: control =D8/D5 0.443

13 Difference in est. death rate (r-c) =D11-D12 -0.093

14 Direction of difference (enter: 1 if D13 is positive or -1 if D13 is negative) -1 -1

16 eq. (V) = Vr = (Ototal*Rr*Rc)/(Rr+Rc)2 =(((D7+D8)*D4*D5)/((D4+D5)^2)) 59.5

17 eq. (VI) = Variance Ln(HR) = 1/Vr =1/D16 0.0168

18 eq. (VII) = Or - Er = (√(Ototal*Rr*Rc)/(Rr+Rc))*�-1(1-p-value/2)*(direction of difference) =(SQRT((D7+D8)*D4*D5)/(D4+D5))*

(NORM.S.INV(1-D9/2)*D14) -22.89

19 eq. (VIII) = ln(HR)=(Or-Er)/Vr =D18/D16 -0.385

21 HR=eLn(HR) =EXP(D19) 0.68

22 95% CI Lower = eLn(HR)-1.96*√(variance Ln(HR)) =EXP(D19-1.96*SQRT(D17)) 0.53

23 95% CI Upper = eLn(HR)+1.96*√(variance Ln(HR)) =EXP(D19+1.96*SQRT(D17)) 0.88

HR=Hazard Ratio, CI=Confidence Interval, eq. = equation, est. = estimated

 

Table 4 Worked Example using Method 4b: Hazard Ratio calculated using median survival estimates

 D F

 Example with equations Example with values

 Raw Data Robotic vs Open Robotic vs Open

4 Total number of patients: robotic 300 300

5 Total number of patients: control 300 300

7 Or (# deaths reported: robotic) 105 105

8 Oc (# deaths reported: control) 133 133

9 MSr (Median Survival): robotic) 3.8 3.8

10 MSc (Median Survival): control) 2.5 2.5

 Calculations 

15 eq. (X) = HR = MSc/MSr =D10/D9 0.66

16 eq. (XI) = Standard Error Ln(HR) = √(1/Or+1/Oc) =SQRT(1/D7+1/D8) 0.13

17 eq. (XII) = CI Lower = ln(HR) - 1.96*SE Ln(HR) =LN(D12)-1.96*D13 -0.67

18 eq. (XII) = CI Lower = ln(HR) + 1.96*SE Ln(HR) =LN(D12)+1.96*D13 -0.16

20 Exponentiate CI Lower = eln(HR) - 1.96*SE Ln(HR) =EXP(D14) 0.51

21 Exponentiate CI Lower = eln(HR) + 1.96*SE Ln(HR) =EXP(D15) 0.85

HR=Hazard Ratio, CI=Confidence Interval, eq. = equation

Question #8

I don’t wish to be snobbish, but the 3D exploded pie chart in Figure 4 seems to me to be a very questionable way of presenting these four values (the information to ink ratio is very low here) and could easily be deleted given that this information is presented in the text.

Response to Question #8:

The figure and all mention of it has been removed from the manuscript.

Question #9

As a final minor comment, I suggest not using “subjects”, as in “when the goal was to maximize the number of included papers and the number of included subjects” (but note at least four other such uses) and instead using “patients” or (if you feel it is appropriate) “participants”.

Response to Question #9:

We have changed all instances of “subjects’ or “subject” to “patients” or “patient” throughout the manuscript.

Reviewer #2: Highly appreciated the author’s effort to consolidate methods for calculating hazard ratio, the most appropriate statistic for a meta-analysis of time dependent outcomes, with clear instructions, a comprehensive worked example, practical tips, and additional calculations. Please address the following concerns

Query 1: Pages 10 to 12: Use of roman numerals to quote equation number prior to the equation. It may be kept after the equation in the right margin for further use

Response to Query #1:

The roman numerals associated with the equations have been moved farthest right.

Query 2: P14, Line 6 : While describing Method 3, the authors specified that “When the

overall p-value seemed to reflect pairwise differences (assumption of similar variances), then the 3-way p-value was assumed to be a reasonable approximation of each pairwise comparison”. This may increase type 1 error rate due to multiple pairwise comparison. How do you address the difference in type I error rate due to each pairwise comparison??

Response to Query #2:

We agree and we have added a statement to the manuscript in the Method 3 section as follows:

 “An additional limitation of this approach is that it may result in type 1 error inflation because the overall comparison is not equivalent to two comparisons to a control.”

Query 3: Decision Tree for Hazard Ratio Extraction specified in Figure 1 is remarkable. However the method 4 mentioned by the authors hold the assumption of no censoring; which is difficult to consider in real research studies. How do you justify this?

Response to Query #3:

Please see response to review 1 question #1 and question #6 regarding Method 4. We agree that an assumption of no censoring is unrealistic and have recommended sensitivity analyses to understand the impact of the use of Method 4. This assumption will lead to the fewest problems when the follow up duration is short, and the censoring mechanism is the same in both arms._________________

---

## [Decision Letter · Decision Letter 1]

9 Dec 2021

PONE-D-21-18912R1Methodology to Standardize Heterogeneous Statistical Data Presentations for Combining Time-to-Event Oncologic OutcomesPLOS ONE

Dear Dr. Slee,

Thank you for submitting your manuscript to PLOS ONE. After careful consideration, we feel that it has merit but does not fully meet PLOS ONE’s publication criteria as it currently stands. Therefore, we invite you to submit a revised version of the manuscript that addresses the points raised during the review process.

We look forward to receiving your revised manuscript.

Kind regards,

Mona Pathak, PhD

Academic Editor

PLOS ONE

Journal Requirements:

Reviewers' comments:

Reviewer's Responses to Questions

**Comments to the Author**

1. If the authors have adequately addressed your comments raised in a previous round of review and you feel that this manuscript is now acceptable for publication, you may indicate that here to bypass the “Comments to the Author” section, enter your conflict of interest statement in the “Confidential to Editor” section, and submit your "Accept" recommendation.

Reviewer #3: (No Response)

2. Is the manuscript technically sound, and do the data support the conclusions?

Reviewer #3: Yes

3. Has the statistical analysis been performed appropriately and rigorously? 

Reviewer #3: Yes

4. Have the authors made all data underlying the findings in their manuscript fully available?

Reviewer #3: No

5. Is the manuscript presented in an intelligible fashion and written in standard English?

Reviewer #3: Yes

6. Review Comments to the Author

Reviewer #3: As someone who's worked on extracting this kind of data for systematic reviews, I was happy to see this outline of different ways to extract survival outcomes from publications.

It was especially nice to see R code for the method of Guyot et al (in the supplement). The worked examples will also certainly be helpful for many researchers doing this sort of data extraction. Would it be possible to include R and / or SAS code for these examples int he supplement? This would further aid researchers in performing their data extraction without mistakes in simple bits of code. For example, I see that the function NORM.S.INV was used, but I'm not sure without looking it up in both excel and R help files which is the corresponding R function.

Finally, in method 1, it is noted that "For these reasons, we prioritized an adjusted HR using the largest sample that adequately addresses confounding (ie. whole patient population over a matched patient cohort when matching decreased the sample size)." This surprised me, as the whole point of matching is to reduce confounding in studies, even while reducing sample size. While the "whole patient population" will of course have a larger sample size, there is no guarantee that an analysis of such data will in fact bring the least biased results, and in fact failing to account for such variables (through matching or adjusting in a regression analysis) would likely lead to biased estimates of the true HR. I would therefore argue, as another reviewer has previously also argued, that priority for HR starts with adjusted or matched analyses, and only if no other data are available, then use the unadjusted HR estimates. The overall goal of a systematic review and possibly included meta-analysis is to obtain a best unbiased estimate of the treatment effect and therefore data extraction for such a SR/MA should also be clearly focused on that. The Cochrane Handbook also appears to prefer adjusted to unadjusted treatment effects, see e.g. https://training.cochrane.org/handbook/current/chapter-06#section-6-3

7. PLOS authors have the option to publish the peer review history of their article (what does this mean?). If published, this will include your full peer review and any attached files.

Reviewer #3: **Yes: **Sarah R Haile

---

## [Author Response · Author response to Decision Letter 1]

5 Jan 2022

Reviewer #3: As someone who's worked on extracting this kind of data for systematic reviews, I was happy to see this outline of different ways to extract survival outcomes from publications.

Reviewer #3 Comment #1: It was especially nice to see R code for the method of Guyot et al (in the supplement). The worked examples will also certainly be helpful for many researchers doing this sort of data extraction. Would it be possible to include R and / or SAS code for these examples in the supplement? This would further aid researchers in performing their data extraction without mistakes in simple bits of code. For example, I see that the function NORM.S.INV was used, but I'm not sure without looking it up in both excel and R help files which is the corresponding R function.

Response to Reviewer #3 Comment #1: We originally performed the HR estimate calculations in excel only; however, we went back and created code de novo in R for all of the equations needed for Methods 1-4, and we are providing this R code in the supplement as appendix S3 and re-naming the tips and tricks appendix as S4. This new appendix is cited in the “worked example” section of the paper as follows:

“We are also providing the R code in Supplemental Appendix S3.”

And in the discussion as follows:

“Our methods add to the list of previously published techniques for analyzing time-to-event outcomes (4, 7,18), and include details for implementing our strategies (Supplemental Appendices – S1 Guyot code, S3 R code for equations of interest, and S4 Data extraction & tricks).”

The addition to the supplement is as follows:

Supplemental Appendix S3: R functions for HR estimate calculations

#Method 1: reverse reference group

 RevRef <- function(HR.Ref1, HRCIL.Ref1, HRCIU.Ref1) {

 #Display for HR (CI) as entered

 DISP1<-paste("HR (CI), Ref. Grp. 1: ", round(HR.Ref1,digits=4), " (", 

 round(HRCIL.Ref1,digits=4), ", ", round(HRCIU.Ref1,digits=4), ")" )

 print(DISP1)

 DISP2<-paste("HR (CI), Ref. Grp. 2: ", round(1/HR.Ref1,digits=4), " (", 

 round(1/HRCIU.Ref1,digits=4), ", ", round(1/HRCIL.Ref1,digits=4), ")" )

 print(DISP2)

 rm(HR.Ref1, HRCIL.Ref1, HRCIU.Ref1, DISP1, DISP2)

 }

 #Example of function call

 #Syntax: RevRef(HR with original ref, HR CI lower limit (orig.), HR CI upper limit (orig.))

 #Example:

 RevRef(1.47, 1.14, 1.90)

#####################

#Method 1: Deriving CI from Reported HR and P-value

 GetCI <- function(HR, Pval) {

 SELnHR <- log(HR)/qnorm((1-Pval/2), mean=0, sd=1)

 HR.CIL <- exp(log(HR) + SELnHR*1.96) 

 HR.CIU <- exp(log(HR) - SELnHR*1.96)

 DISP1 <- paste("HR (CI), p-value: ", round(HR,digits=4), " (", round(HR.CIL,digits=4), 

 ", ", round(HR.CIU,digits=4), "), p = ", round(Pval,digits=4))

 print(DISP1)

 rm(HR, Pval, SELnHR, HR.CIL, HR.CIU, DISP1)

 }

 #Example of function call

 #Syntax: GetCI(Hazard Ratio, Log-Rank p-value)

 #Example:

 GetCI(.68, 0.0032)

#####################

#Method 2: Hazard Ratio as calculated using event counts

 M2EvCt <- function(Rr, Rc, Or, Oc, Pval) {

 Dth.r <- Or / Rr

 Dth.c <- Oc / Rc

 Dth.Diff = Dth.r - Dth.c

 Pmult <- ifelse(Dth.Diff < 0, -1, 1)

 inv.Vr <- 1/(((Or + Oc)*Rr*Rc)/((Rr + Rc)^2))

 O_E <- (sqrt((Or + Oc)*Rr*Rc)/(Rr + Rc)*(qnorm((1-Pval/2), mean=0, sd=1))*Pmult)

 LnHR <- O_E*inv.Vr

 HR <- exp(LnHR)

 HR.CIL <- exp (LnHR - 1.96*sqrt(inv.Vr))

 HR.CIU <- exp (LnHR + 1.96*sqrt(inv.Vr))

 DISP1 <- paste("HR (CI), p-value: ", round(HR,digits=4), " (", round(HR.CIL,digits=4), 

 ", ", round(HR.CIU,digits=4), "), p = ", round(Pval,digits=4))

 print(DISP1)

 rm(Rr, Rc, Or, Oc, Pval, Dth.r, Dth.c, Dth.Diff, Pmult, inv.Vr, O_E, LnHR, HR, HR.CIL, HR.CIU, DISP1)

 }

 #Example of function call

 #Syntax: M2EvCt(Total # group 1 patients , Total # group 2 patients, # events group 1, 

 # events group 2, Log-Rank p-value)

 #Example:

 M2EvCt(300, 300, 105, 133, 0.003)

#####################

#Method 4a: Hazard Ratio as calculated using KM estimates

 M4aHR.KM <- function(Rr, Rc, Surv.KMr, Surv.KMc, Pval) {

 Or <- (Rr - (Rr*Surv.KMr))

 Oc <- (Rc - (Rc*Surv.KMc))

 Dth.r <- Or / Rr

 Dth.c <- Oc / Rc

 Dth.Diff = Dth.r - Dth.c

 Pmult <- ifelse(Dth.Diff < 0, -1, 1)

 inv.Vr <- 1/(((Or + Oc)*Rr*Rc)/((Rr + Rc)^2))

 O_E <- (sqrt((Or + Oc)*Rr*Rc)/(Rr + Rc)*(qnorm((1-Pval/2), mean=0, sd=1))*Pmult)

 LnHR <- O_E*inv.Vr

 HR <- exp(LnHR)

 HR.CIL <- exp (LnHR - 1.96*sqrt(inv.Vr))

 HR.CIU <- exp (LnHR + 1.96*sqrt(inv.Vr))

 DISP1 <- paste("HR (CI), p-value: ", round(HR,digits=4), " (", round(HR.CIL,digits=4), 

 ", ", round(HR.CIU,digits=4), "), p = ", round(Pval,digits=4))

 print(DISP1)

 rm(Rr, Rc, Or, Oc, Surv.KMr, Surv.KMc, Pval, Dth.r, Dth.c, Dth.Diff, Pmult,

 inv.Vr, O_E, LnHR, HR, HR.CIL, HR.CIU, DISP1)

 }

 #Example of function call

 #Syntax: M4aHR.KM(Total # group 1 patients , Total # group 2 patients, KM Survival group 1, 

 KM Survival group 2, Log-Rank p-value)

 #Example:

 M4aHR.KM(300, 300, 0.586, 0.445, 0.003)

#####################

#Method 4b. Hazard Ratio calculated using median survival estimates

 M4bMedSurv <- function(Or, Oc, MSr, MSc) {

 HR <- MSc/MSr

 SELnHR <- sqrt((1/Or)+(1/Oc))

 LnHR <- log(HR)

 HR.CIL <- exp(LnHR - 1.96*SELnHR)

 HR.CIU <- exp(LnHR + 1.96*SELnHR)

 DISP1 <- paste("HR (CI): ", round(HR,digits=4), " (", round(HR.CIL,digits=4), 

 ", ", round(HR.CIU,digits=4),")")

 print(DISP1)

 rm(Or, Oc, MSr, MSc, SELnHR, LnHR, HR, HR.CIL, HR.CIU, DISP1)

 }

 #Example of a function call

 #Syntax: M4bMedSurv(# events group 1, # events group 2, Median survival for group 1, Median survival for group 2)

 #Example:

 M4bMedSurv(105, 133, 3.8, 2.5) 

Reviewer #3 Comment #2: Finally, in method 1, it is noted that "For these reasons, we prioritized an adjusted HR using the largest sample that adequately addresses confounding (ie. whole patient population over a matched patient cohort when matching decreased the sample size)." This surprised me, as the whole point of matching is to reduce confounding in studies, even while reducing sample size. While the "whole patient population" will of course have a larger sample size, there is no guarantee that an analysis of such data will in fact bring the least biased results, and in fact failing to account for such variables (through matching or adjusting in a regression analysis) would likely lead to biased estimates of the true HR. I would therefore argue, as another reviewer has previously also argued, that priority for HR starts with adjusted or matched analyses, and only if no other data are available, then use the unadjusted HR estimates. The overall goal of a systematic review and possibly included meta-analysis is to obtain a best unbiased estimate of the treatment effect and therefore data extraction for such a SR/MA should also be clearly focused on that. The Cochrane Handbook also appears to prefer adjusted to unadjusted treatment effects, see e.g. https://training.cochrane.org/handbook/current/chapter-06#section-6-3

Response to Reviewer #3 Comment #2: We agree that matched or adjusted HR should be used preferentially over an unadjusted HR. We apologize for the lack of clarity in our statement. Our comment was meant to address the case when a paper reported both an adequately adjusted HR and a matched HR for the same data set. To clarify, we have modified the statement in the “Method 1” section as follows:

"When multiple hazard ratios were reported, the statistical analysis that produced the hazard ratio was also captured (i.e., univariable, multivariable) to follow the extraction priority. It is important to determine an extraction preference a priori for when more than one hazard ratio is reported. Our criterion was to prioritize adjusted or matched analyses over unadjusted data, and when both adjusted and matched analyses were available, to maximize group size, because analyses using entire populations account for the relative frequency of case types, severity of disease, surgeon experience, etc. and the results are more generalizable. For these reasons, we prioritized an adjusted HR using the largest sample that adequately addresses confounding (ie. adjusted analysis using the whole patient population over a matched patient cohort when matching decreased the sample size)."

We also clarified that an adjusted or matched analysis is preferable to an unmatched analysis by modifying the Quality Control Methods section of the manuscript using wording similar to that of the reviewer, as follows: 

“When more than one analysis was reported, we selected an adjusted or matched analysis preferentially, and only if no other data were available, unadjusted data; we chose the largest available analysis that adequately addressed cohort differences using the same hierarchy as listed above. When no adjustment or matching was performed, the comparability of the groups was determined by comparing baseline values for a list of covariates potentially related to oncologic outcomes.”

We also clarified this priority in supplemental appendix S4: Assumptions, Rules, and Tips as follows:

“Our Rule: Use an adjusted or matched HR over an unmatched HR. In instances where both an adjusted and a matched HR are provided, to maximize group size, use an adjusted HR using the whole patient population over a matched patient cohort when matching decreased the sample size.”

Journal Requirements - References: Please review your reference list to ensure that it is complete and correct. If you have cited papers that have been retracted, please include the rationale for doing so in the manuscript text, or remove these references and replace them with relevant current references. Any changes to the reference list should be mentioned in the rebuttal letter that accompanies your revised manuscript. If you need to cite a retracted article, indicate the article’s retracted status in the References list and also include a citation and full reference for the retraction notice.

Response to Journal Requirements: All references have been checked for completeness, and citations have been modified to add details as shown in the tracked changes version of the manuscript. There are no retracted references; however, the Parmar 1998 reference had an Erratum/correction published that corrected one of the equations that was not relevant for our work, and those details have been added to the end of the reference as follows:

“Parmar MKB, Torri V, Stewart L. Extracting summary statistics to perform meta‐analyses of the published literature for survival endpoints. Statistics in medicine. 1998;17(24):2815-34. Corrected: Stat Med. 2004 Jun 15;23(11):1817.”

We have confirmed that all references can be found by searching Pubmed or google using the citations as currently listed.

There was a glitch in our reference handling software; therefore, the following references have been corrected as follows:

Reference #2 has been corrected to: 

“2. Tewari A, Sooriakumaran P, Bloch DA, Seshadri-Kreaden U, Hebert AE, Wiklund P. Positive surgical margin and perioperative complication rates of primary surgical treatments for prostate cancer: a systematic review and meta-analysis comparing retropubic, laparoscopic, and robotic prostatectomy. Eur Urol. 2012;62(1):1-15.”

Reference #5 has been corrected to:

“5. Williamson PR, Smith CT, Hutton JL, Marson AG. Aggregate data meta-analysis with time-to-event outcomes. Stat Med 2002;21:3337-51.”

The order of Ref #7 Parmar 1998 and Ref #8 Higgins Chapter 6 has been swapped.

Reference #12 for the Cochrane Handbook has been updated to the latest online version:

“Ref #12 Higgins JPT, Thomas J, Chandler J, Cumpston M, Li T, Page MJ, et al. Cochrane handbook for systematic reviews of interventions. 6.2 ed (updated February 2021): Cochrane, 2021. Available from www.training.cochrane.org/handbook.”

Ref #18 has been removed and references 19-24 have been re-numbered to references 18-23.

Guidelines for resubmitting figure files: While revising your submission, please upload your figure files to the Preflight Analysis and Conversion Engine (PACE) digital diagnostic tool, https://pacev2.apexcovantage.com/. PACE helps ensure that figures meet PLOS requirements. To use PACE, you must first register as a user. Registration is free. Then, login and navigate to the UPLOAD tab, where you will find detailed instructions on how to use the tool. If you encounter any issues or have any questions when using PACE, please email PLOS at figures@plos.org. Please note that Supporting Information files do not need this step.

Response to Guidelines for resubmitting figure files: The three figures to be included in the main body of the manuscript have been run through the PACE tool. The PACE tool changed the resolution to 300 ppi, converted them to a valid TIF file, and changed the names to meet requirements.

---

## [Decision Letter · Decision Letter 2]

25 Jan 2022

Methodology to Standardize Heterogeneous Statistical Data Presentations for Combining Time-to-Event Oncologic Outcomes

PONE-D-21-18912R2

Dear Dr. Slee,

We’re pleased to inform you that your manuscript has been judged scientifically suitable for publication after incorporation of suggestions by reviewer and will be formally accepted for publication once it meets all outstanding technical requirements.

Kind regards,

Mona Pathak, PhD

Academic Editor

PLOS ONE

Reviewers' comments:

Reviewer's Responses to Questions

**Comments to the Author**

1. If the authors have adequately addressed your comments raised in a previous round of review and you feel that this manuscript is now acceptable for publication, you may indicate that here to bypass the “Comments to the Author” section, enter your conflict of interest statement in the “Confidential to Editor” section, and submit your "Accept" recommendation.

Reviewer #3: All comments have been addressed

2. Is the manuscript technically sound, and do the data support the conclusions?

Reviewer #3: Yes

3. Has the statistical analysis been performed appropriately and rigorously? 

Reviewer #3: Yes

4. Have the authors made all data underlying the findings in their manuscript fully available?

Reviewer #3: No

5. Is the manuscript presented in an intelligible fashion and written in standard English?

Reviewer #3: Yes

6. Review Comments to the Author

Reviewer #3: Thank you for addressing my comments, and for adding the code.

It would be more readable for readers if the code did not round all of the numbers, and instead printed them out as usual. I would also point out that it is not necessary to rm() objects at the end of a function, as R does not keep things created during a function call. See some example code I adapted for RevRef and GetCI below, though the other functions should be adapted with readability in mind, that is so that readers see what the function is doing without worrying about how to round or print the numbers.

RevRef <- function(HR.Ref1, HRCIL.Ref1, HRCIU.Ref1) {

Hazard.Ratio <- c(HR.Ref1, 1/ HR.Ref1)

lb <- c(HRCIL.Ref1, 1 / 1/HRCIU.Ref1)

ub <- c(HRCIU.Ref1, 1 / 1/HRCIL.Ref1)

data.frame(reference = c("Group 1", "Group 2"),

Hazard.Ratio, lb, ub)

}

RevRef(1.47, 1.14, 1.90)

#Method 1: Deriving CI from Reported HR and P-value

GetCI <- function(HR, Pval) {

SELnHR <- log(HR)/qnorm((1-Pval/2), mean=0, sd=1)

HR.CIL <- exp(log(HR) + SELnHR*1.96)

HR.CIU <- exp(log(HR) - SELnHR*1.96)

c("HR" = HR, "lb" = HR.CIL, "ub" = HR.CIU, p.value = Pval)

}

GetCI(.68, 0.0032)

7. PLOS authors have the option to publish the peer review history of their article (what does this mean?). If published, this will include your full peer review and any attached files.

Reviewer #3: **Yes: **Sarah R Haile

---

## [Editor Report · Acceptance letter]

10 Feb 2022

PONE-D-21-18912R2 

Methodology to Standardize Heterogeneous Statistical Data Presentations for Combining Time-to-Event Oncologic Outcomes    

Dear Dr. Slee:

I'm pleased to inform you that your manuscript has been deemed suitable for publication in PLOS ONE. Congratulations! Your manuscript is now with our production department. 

Kind regards, 

on behalf of

Dr. Mona Pathak 

Academic Editor

PLOS ONE